# Epstein–Barr Virus—Associated Malignancies and Immune Escape: The Role of the Tumor Microenvironment and Tumor Cell Evasion Strategies

**DOI:** 10.3390/cancers13205189

**Published:** 2021-10-16

**Authors:** Marcus Bauer, Simon Jasinski-Bergner, Ofer Mandelboim, Claudia Wickenhauser, Barbara Seliger

**Affiliations:** 1Department of Pathology, Martin Luther University Halle-Wittenberg, Magdeburger Str. 14, 06112 Halle (Saale), Germany; marcus.bauer@uk-halle.de (M.B.); claudia.wickenhauser@uk-halle.de (C.W.); 2Department of Medical Immunology, Martin Luther University Halle-Wittenberg, Magdeburger Str. 2, 06112 Halle (Saale), Germany; simon.jasinski@uk-halle.de; 3Department of Immunology, Faculty of Medicine, The Hebrew University of Jerusalem, En Kerem, P.O. Box 12271, Jerusalem 91120, Israel; oferm@ekmd.huji.ac.il; 4Fraunhofer Institute for Cell Therapy and Immunology, Perlickstr. 1, 04103 Leipzig, Germany

**Keywords:** EBV, EBV-associated malignancies, malignant transformation, tumor microenvironment, immune escape

## Abstract

**Simple Summary:**

The Epstein–Barr virus, also termed human herpes virus 4, is a human pathogenic double-stranded DNA virus. It is highly prevalent and has been linked to the development of 1–2% of cancers worldwide. EBV-associated malignancies encompass various structural and epigenetic alterations. In addition, EBV-encoded gene products and microRNAs interfere with innate and adaptive immunity and modulate the tumor microenvironment. This review provides an overview of the characteristic features of EBV with a focus on the intrinsic and extrinsic immune evasion strategies, which contribute to EBV-associated malignancies.

**Abstract:**

The detailed mechanisms of Epstein–Barr virus (EBV) infection in the initiation and progression of EBV-associated malignancies are not yet completely understood. During the last years, new insights into the mechanisms of malignant transformation of EBV-infected cells including somatic mutations and epigenetic modifications, their impact on the microenvironment and resulting unique immune signatures related to immune system functional status and immune escape strategies have been reported. In this context, there exists increasing evidence that EBV-infected tumor cells can influence the tumor microenvironment to their own benefit by establishing an immune-suppressive surrounding. The identified mechanisms include EBV gene integration and latent expression of EBV-infection-triggered cytokines by tumor and/or bystander cells, e.g., cancer-associated fibroblasts with effects on the composition and spatial distribution of the immune cell subpopulations next to the infected cells, stroma constituents and extracellular vesicles. This review summarizes (i) the typical stages of the viral life cycle and EBV-associated transformation, (ii) strategies to detect EBV genome and activity and to differentiate various latency types, (iii) the role of the tumor microenvironment in EBV-associated malignancies, (iv) the different immune escape mechanisms and (v) their clinical relevance. This gained information will enhance the development of therapies against EBV-mediated diseases to improve patient outcome.

## 1. Introduction

The Epstein–Barr virus (EBV), described in 1964 by Michael Anthony Epstein and Yvonne Barr in African endemic Burkitt lymphoma (BL) samples, was the first discovered human-tumor-associated virus [1,2]. This knowledge provided important insights into the involvement of viruses in the pathogenesis of human malignancies and the natural history of human herpes viruses (HHVs). EBV, also known as human herpes virus 4 (HHV4), a family member of the gamma herpesviridae, is characterized by a high transmission rate and a spread of infection in more than 90% of the world population [3]. The virus consists of a 170–180 kb linear double-stranded (ds) enveloped DNA with a toroid-shaped protein core, a nucleocapsid with 162 capsomers and external virus-encoded glycoprotein spikes on the surface of the viral tegument [4,5,6]. The viral genome encodes more than 85 genes, which, to a distinct extent, contribute to the mechanisms of EBV infection and to the initiation and clinical manifestation of EBV-associated human diseases. For 30 to 40% of EBV genes, very little is known concerning their specific function [7]. So far, two major EBV types, named type 1 and type 2 EBV, with nearly identical genomes except for the genes encoding some of the nuclear proteins, the EBV nuclear antigen (EBNA)-2 and EBNA-3A, -3B and -3C, have been identified [8,9]. Type 1 EBV is found ubiquitously worldwide and has a higher transformation efficiency of B cells, while type 2 EBV is mainly detected in Africa [10,11]. Primary infection mostly involves asymptomatic children prior to the age of 5 years and is rather rare in adults who more frequently acquire severe symptoms called infectious mononucleosis [12]. After the primary infection, the virus shows a life-long persistence in memory B cells [13]. In general, EBV establishes different life-cycle programs consisting of the primary infection, the latency and the lytic program [2]. Within the lytic program of infection, new infectious virions are produced, while the latent form of infection allows the virus to persist in host cells [14]. The majority of EBV-infected individuals control the infection by cytotoxic immune cell responses via natural killer (NK) cells and CD8^+^ T lymphocytes [3,15]. Only a small number of infected individuals develop chronic EBV-associated pathologies, which are often related to immune deficiencies, genetic pre-disposition and environmental factors [16]. Chronic EBV infection of different tissues of mainly epithelial and lymphocytic origin has been associated with malignant diseases such as carcinomas, lymphomas/lymphoproliferative disorders and soft-tissue tumors [17,18,19,20,21]. Geographically, EBV-associated neoplasia is present worldwide but is more frequent in Asia and Africa compared to the Western world [17,22,23,24,25] with a higher incidence in males than in females [2,26]. Furthermore, different EBV-associated tumor entities present distinct peaks in the disease onset. In detail, endemic BL is primarily a disease of young children between 2 and 20 years of age. In contrast, EBV-positive nasopharyngeal carcinoma (NPC) and gastric carcinoma (GC) occur in adults between 18 and 80 years with EBV-positive GC patients frequently younger than 60 years [17,18,20,26,27,28]. A comprehensive summary concerning the geographic and epidemiologic peculiarities of EBV-associated malignancies is shown in Table 1.

## 2. Viral Life Cycle and EBV-Associated Transformation

EBV infections have a high prevalence in the world population of more than 90% and show a life-long latency in the host. The primary EBV infection occurs via orally transmitted virions infecting resting B cells and the oral epithelium, although it remains largely unclear whether the B cells or the epithelial cells are infected first [2,47]. While the exact mechanism of viral entry in these epithelial host cells is under discussion, the viral entry into naïve B cells is initiated by attachment of the EBV glycoprotein (gp) gp350 to the CD21 surface molecule known as complement C3d receptor 2 (CR2) that is selectively expressed on lymphoid cells [48]. For persistent EBV infection, the virus has to enter the circulating memory B-cell pool as reviewed elsewhere [2,49]. Within this process, EBV shows different latency types that are characterized by a distinct expression pattern of a limited number of EBV genes. Within the circulating memory B cells, the EBV infection passes into the most restricted latency type 0. This latency type is characterized by the sole expression of non-coding genes, including EBER transcripts, BARTs (BamHI fragment A rightward transcript) and several microRNAs (miRs) [50]. Three further latency types can be distinguished, which are characterized by the differential expression of five EBV-encoded nuclear antigens (EBNAs), two latent membrane proteins (LMPs), two EBV-encoded small RNAs (EBERs) and non-coding BART RNAs [51]. A detailed summary of the distinct gene expression profiles observed in the latency phase types 0, I, II and III of EBV infection has been recently reviewed and is provided in Table 2 [51,52,53].

Furthermore, a lytic EBV infection, which occurs in both immunocompetent and immuno-incompetent hosts as part of the primary infection of the oropharynx, can be distinguished from the latent infection types [54]. Three different lytic phases have been described, namely immediate early, early and late phases, which are associated with the expression of more than 80 lytic genes [50]. Within lytic infection, host cells are destroyed and new infectious virions are released. The expression of the immediate early transcription factor Zta, which is encoded in the BZLF1 gene and Rta (BRLF1 gene product), initiates this lytic phase [55,56,57].

Since the first discovery of EBV in African endemic BL samples in 1964, the malignant transformation capacity of EBV-infected cells has been extensively investigated. Interestingly, despite an infection rate of over 90%, only 1.8% of worldwide cancer deaths can be attributed to EBV-associated malignancies [58,59]. EBV-associated malignancies express different EBV latent gene products with oncogenic potential that help to distinguish between distinct entities in the diagnostic setting [60]. The EBV gene products of BALF1 and BHRF1, which are BCL-2 homologs with anti-apoptotic functions, are both known to be involved in B-cell transformation [61,62]. Furthermore, the EBV latent gene products interfere with the innate and adaptive immunity by modulating the tumor microenvironment (TME) and thereby supporting tumor progression, which is discussed in detail in Section 4.3 and Section 4.4. In addition, EBV-encoded miRs such as miR-BHRF1-1, miR-BHRF1-2 and miR-BHRF1-3 [63] are expressed during EBV latency type III infection and the lytic phase. These miRs are also known to inhibit apoptosis and enhance cell-cycle progression in the early phase of B-cell infection [62]. MiRs are small 19–24-nucleotide-long non-coding single-stranded RNAs involved in the post-transcriptional gene regulation, preferentially but not exclusively binding to the 3′ untranslated region (UTR) of their targets, leading to a translational inhibition and mRNA decay or mRNA storage [64,65,66]. MiRs are usually expressed in eukaryotes, but certain DNA viruses, including herpes viruses, can also encode for miRs, which potentiate their transforming properties and mediate immune escape mechanisms [57,67,68]. Recently, the role of non-coding RNAs of EBV with a focus on EBERs and miRs in EBV-mediated tumorigenesis and immune control has been extensively reviewed and is referred to for additional information [53]. Furthermore, an influence of lytic gene expression in the process of malignant transformation has been shown [57].

## 3. Identification of EBV-Infected Cells in Their Tissue Context, Its Activity and Diagnostic or Therapeutic Approaches in Oncology

Since latent EBV infection has been implicated in the pathogenesis of diverse malignancies, suitable test methods to evaluate gene integration and gene replication activity in EBV-infected cells are necessary to design proper biomarkers for diagnosis, disease progression and monitoring of treatment. Thus, the analysis of distinct EBV genomic regions, activation-associated gene products/proteins and antibody production following EBV infection is mandatory. Over the last decades, a number of different methods to identify EBV-related gene products and proteins in both blood and formalin-fixed, paraffin-embedded (FFPE) tissue samples have been developed, which differ in terms of their significance and include serologic and molecular studies of EBV-associated gene products that predetermine the choice of the suitable detection method and are related to the clinical context [60,69]. It is noteworthy that the expression levels of certain EBV-specific genes determine the different latency types and are distinct in EBV-associated malignancies as summarized in Table 2.

### 3.1. EBV-Encoded RNA Detection

EBV-encoded RNA (EBER) in situ hybridization (ISH) is the “gold standard” for detecting and localizing EBV-infected cells in biopsy samples known to be the most sensitive method since EBERs are consistently expressed in all latent EBV infection types independent of their origin from neoplastic or morphologically normal tissues [60,70,71]. Although the application is widespread, there exist some limitations. While false-negative EBER hybridizations may result as a consequence of RNA degradation [72], false-positive EBER results may be attributable to latent infection of background lymphocytes or artifacts, such as non-specific staining or cross-reactivity with mucin, yeast or plant materials [72,73]. Thus, confirming the results by other diagnostic tools next to EBER-ISH increases the diagnostic accuracy.

### 3.2. EBV-Specific DNA Detection

Real-time polymerase chain reaction (PCR) is a fast and sensitive widely used method for evaluation of the virus load [74,75]. In detail, the quantification of episomal EBV DNA coding for EBNA-1 and the viral envelop glycoprotein (gp) 220 in the plasma and peripheral blood mononuclear cells (PBMCs) had to stand the test as a suitable marker of acute EBV infection that correlates with clinical symptoms [76,77]. In FFPE tissues, similar sensitivity and reliability of this method compared to EBER-ISH were shown, suggesting its use for EBV diagnostic screening [78]. It has to be taken into account that the viral load is higher in the oral cavity than in peripheral blood [79]. Considering this restriction as a quantitative tool, evaluation of the virus load helps to determine the severity of an EBV infection/reactivation and to identify patients with EBV-infection-related diseases. However, false-positive results in blood samples can be achieved due to the detection of EBV-positive memory cells, and therefore this method is not suitable to verify EBV-associated malignancies [78,80,81,82].

### 3.3. Detection of EBV-Associated Biomarkers

Latent EBV infections differ by their expression pattern of virus-coded proteins, which provides a diagnostic tool. Latency type 0 represents an antigen-negative form of infection only expressing EBER and BART miRs, while latency type I shows a selective expression of EBNA1. In latency type II, an expression of LMP-1, LMP-2A and LMP-2B is detectable. In addition, latency type III involves the expression of all six EBNA proteins [2,60,73,83]. Evaluation of the expression patterns in FFPE tissue samples can be easily assessed by immunohistochemistry (IHC) and ISH. In this context, evaluation of LMP-1 and EBNA-2 expression is an extremely simple and cost-effective tool to determine the latency types as demonstrated in Figure 1.

### 3.4. Detection of EBV-Specific Antibodies

To evaluate the strength of the interaction between the virus and the host, various methods are available for serologic detection of antibodies against different structures of EBV. As a fast primary screening method, the heterophile antibody test detects subsets of antibodies produced by the human immune system in response to EBV infection. Specific immunologic tests include, e.g., enzyme-linked immunosorbent assay (ELISA), immunofluorescence assay, Western blot and IgG avidity assay as well as multiplex flow immunoassay [84,85]. Although displaying a high degree of variability, it is generally possible with this repertoire to define the individual infection status and to allow the distinction between acute, latent and reactivated infection. For screening, the viral capsid antigen (VCA) IgG, VCA IgM and EBNA-1 IgG are employed. The presence of VCA IgM and VCA IgG without EBNA-1 IgG correlates with acute EBV infection, whereas the presence of VCA IgG and EBNA-1 IgG without VCA IgM indicates a past infection. However, the interpretation of the serological findings can be difficult, since VCA IgG can be present without VCA IgM or EBNA-1 IgG in acute or past infection. In addition, all three parameters may be simultaneously detected upon acute or latent EBV infection [84]. No significant relationship between EBV serology and the presence of EBV in Hodgkin–Reed/Sternberg (HRS) was detectable in classical Hodgkin lymphoma (cHL) [86].

### 3.5. Extracellular Vesicles (EVs) as Biomarkers for EBV-Associated Diseases

Extracellular vesicles (EVs) are nanoparticles that can be released from EBV-infected cells and are membrane-surrounded structures. They are stratified by their size or their mechanisms of biogenesis and are classified into exosomes (30–150 nm), microvesicles (100 nm–1 µm) and apoptotic bodies (1–5 µm). EVs represent key factors of the intercellular cell–cell communication through the delivery of biologically active cargo containing a plethora of proteins, lipids, nucleic acids (e.g., miRs, lncRNAs, circRNA, DNA) and metabolites that can be taken up by distant cell types thereby affecting physiologic and pathophysiologic processes [87]. Thus, EVs are major players in cell growth, invasion, angiogenesis and immune cell regulation, which contribute to the development and progression of malignancies [88].

Recent advances in methods have facilitated the isolation of EVs, which are therefore in particular candidate biomarkers for liquid biopsies [89,90]. In this context, it was reported that EVs composed of different viral components including viral miRs (miR-BARTs) and proteins are released from EBV-infected cells, which have profound effects on the cellular microenvironment [91]. Based on their cargo, EBV EVs play an important role in the regulation of EBV infection and the pathogenesis of EBV-associated diseases. The EBV oncoproteins LMP-1 and BARF1 were detected in EVs in the serum and saliva of NPC patients [92], which are also candidate biomarkers for NK/T cell lymphoma [93]. Thus, EVs might be helpful tools for diagnosis and prognosis as well as therapeutic targets in EBV-associated malignancies [94,95]. This is underlined by many efforts to create technical solutions to track EVs/exosomes with traceable markers for their use as potential biomarkers.

## 4. Mechanisms of Latent-EBV-Infection-Induced Malignancies

There exists a broad spectrum of malignancies following chronic latent EBV infection encompassing a couple of different cell types and anatomical localizations. For the distinct entities, EBV-associated versus non-EBV-associated malignancies differ regarding their gene expression profiles, metabolism, signal transduction and their immune escape mechanisms and the consecutive composition of the different players of the tumor microenvironment (TME) [52].

### 4.1. Somatic Mutations

In recent years, numerous genomic studies have been carried out demonstrating a higher mutational burden in EBV-positive compared to EBV-negative malignancies, which was even higher in type 1 compared to type 2 latency types [96]. It has been suggested that latent EBV gene and protein expression may contribute to genome instability in these tumors. In this context, LMP1 is known to impair the mitotic G2 checkpoint and lead to chromosomal instability by accumulation of somatic mutations [97], while other studies indicated that EBNA-1 may promote genomic instability [98]. EBNA-1 has been shown to act as a transcription factor thereby increasing the expression of cellular genes potentially important for oncogenesis [99]. Furthermore, EBNA-1 has been shown to promote DNA damage and genomic instability due to the generation of reactive oxygen species (ROS) [100]. Entity-independent somatic mutations following EBV infection and somatic mutations specific for EBV-associated hematologic malignancies or carcinoma are summarized in Table 3. As an example of entity-dependent mutations, alterations in the PI3K/AKT/MAPK signaling pathway, in particular in PIK3CA, are detected in EBV-associated NPC and GC but not in EBV-associated lymphoma [101,102,103].

In contrast, inactivating mutations or decreased protein expression of ARID1A encoding a member of chromatin-remodeling proteins were detected in both EBV-associated carcinoma and lymphoma [102,104,105]. However, it is noteworthy that ARID1A aberrations were reported in EBV-associated as well as in EBV-negative GC subtypes [106]. Lower frequencies of TP53 mutations that usually show a positive correlation with higher mutational burden have been reported for EBV-positive NPC, GC, BL and PTLD [105,107,108,109,110,111] despite that the p53 pathway is frequently deregulated in these diseases [110]. As an exception, no correlation between TP53 mutations and EBV status was found in cHL [112]. Deciphering the mutational landscape of EBV-associated malignancies provided new insights into their tumorigenesis and elucidated the mechanisms of how EBV-induced malignancies manipulate the immune system. These issues are described in Section 4.3 and Section 4.4, respectively.

**Table 3 cancers-13-05189-t003:** Genetic landscape of EBV-associated malignancies. (↓: down in EBV-associated malignancies, ↑: up in EBV-associated malignancies).

Function	Gene	Type of Genetic Aberration	Frequency in EBV-Associated Malignancies	EBV-Associated Malignancies	References
DNA repair	TP53	Inactivation	↓	NPC, GC, BL, PTLD	[104,105,109,110,111,113]
Signal transduction	PIK3CA	Inactivation	↑	NPC, GC	[101,102,103]
	PIK3R1	Inactivation	↑	GC	[102]
	SMAD4	Inactivation	↑	GC	[102]
Chromatin remodeling	ARID1A	Inactivation	↑	NPC, GC,	[102,104,105]
Transcription factor	MYC	Activation	↓	BL	[105]
IFN signaling	JAK2	Amplification	↑	GC	[103]
	SOCS1	Mutation	↓	GC, PTLD	[111]
NF-ĸB pathway	TRAF3	Inactivation	↑	NPC	[114,115,116]
	CYLD	Inactivation	↑	NPC	[114,115,116]
	NF-ĸBIA	Inactivation	↑	NPC	[114,115,116]
Antigen presentation	MHC-I	Inactivation	↑	NPC	[116]

### 4.2. Epigenetic Alterations

EBV infection is an epigenetic driver and massively alters the gene signature and gene regulation in infected host cells. The main mechanisms in EBV-associated malignancies encompass altered DNA methylation and histone acetylation.

DNA hypermethylation of genes has been described in various diseases [52]. Global DNA methylation changes toward increased CpG island hypermethylation were found in human immortalized normal oral keratinocytes after EBV infection [117] and also in EBV-associated malignancies [118]. Thus, DNA methylation affects both the host and the EBV genome. The expression of different virus-encoded genes in infected host cells is strictly controlled by DNA methylation of viral promoter CpG islands, which is a prerequisite for the different phases of the viral life cycle [119]. Genes predominantly involved in the lytic phase are regulated by promoter hypermethylation leading to their repression. The targeting of such epigenetically repressed promoter regions by BZLF1 gene product Zta reverses their silencing and enhances the expression of genes required for the lytic phase [119]. Next to EBV-positive GC, exhibition of an extremely high DNA methylation pattern was also shown in EBV-positive lymphoma and NPC specimens. NPC-derived cell lines exert hypermethylation of the EBV transcription start sites when compared to non-malignant corresponding EBV-infected human tissues [120]. These data are in line with a study comparing seven EBV-positive NPC lesions and five non-cancer nasopharyngeal epithelium tissues regarding the methylation status of seven candidate genes with known hypermethylated promoter CpG islands and reduced expression in NPC tissues, such as CR2, ITGA4, RERG, RRAD, SHISA3, ZNF549 and ZNF671, demonstrating significantly higher methylation rates of these genes in NPC than in control tissues [114]. In contrast, a clinical study of EBV-positive and EBV-negative HL patients investigating a panel of seven selected tumor-associated human genes known to be hypermethylated in various malignancies (RASSF1A, P16, CDH1, DAPK, GSTP1, SHP1 and MGMT) demonstrated that the promoters of these genes were more frequently hypermethylated in EBV-negative than in EBV-positive cases [121]. Furthermore, there might exist a competition of DNA methylation between different host DNA sequences and EBV DNA sequences. This is underlined by the fact that the EBV genome could be a target of the human DNA methyl-transferases (DNMTs) since the RNAi-mediated DNMT1 and DNMT3B depletion resulted in a hypomethylation of CpG sites in the EBV genome [120]. So far, only a limited number of in vitro and in vivo studies with small cohort sizes and even using different methods for DNA methylation analyses are available.

Another important epigenetic mechanism regulating gene expression is the alteration of histone acetylation. Histone acetylation enables gene transcription by assessing the DNA locus via changes in the chromatin structure, whereas histone deacetylation leads to the suppression of gene transcription [122]. In an in vitro study, the treatment of Raji cells with the histone deacetylase inhibitor trichostatin A resulted in a minor BZLF1 induction [123]. However, distinct EBV-positive cell lines exhibited varying sensitivities when treated with different histone deacetylase inhibitors resulting in a lytic activation induced by BZLF1 and BRLF1 [124].

### 4.3. Tumor Microenvironment

The TME plays a central role in local cancer control by recruiting and differentiating immune-suppressive and/or anti-inflammatory cells, such as regulatory T cells (Tregs), Th17 cells, dendritic cells (DCs), M2 tumor-associated macrophages (TAMs) and myeloid-derived suppressor cells (MDSCs), and inhibiting immune effector cells such as NK cells and CD8^+^ T lymphocytes, which leads to the establishment of an immunosuppressive TME [125]. Next to neoplastic properties, it has been known for decades that EBV infection can influence the composition and function of the TME consisting of both innate and adaptive immune cells, different soluble factors and EVs, which depend on the EBV-associated malignancies (Figure 2) [126,127,128,129,130]. In EBV-driven malignancies, the TME is modulated for viral benefit, thereby affecting disease progression. After infection, EBV can regulate its own viral and non-viral protein expression within the host cell and, in case of malignancies, actively modulate the tumor phenotype and in turn the tumor/TME interaction [131]. Although highly variable, the density of lymphocytes and plasma cells within the tumor stroma and of EBV-associated malignancies is elevated when compared to EBV-negative neoplasia as seen in other virus-associated tumors [132,133]. Furthermore, multiple viral infections within the same tumors could influence the TME [105]. However, some characteristics are more common between EBV-associated tumors compared to EBV-negative counterparts, which do not depend on the anatomical localization or cellular origin [23,114,126,127,128,129,130,134,135]. It is noteworthy that for some EBV-associated diseases only a little information on the TME is available due to their low incidence, such as NK/T cell lymphoma. Over the last years, the TME of EBV-positive NPC and GC was extensively investigated and demonstrated an unusual lymphocyte-rich stroma [136,137,138]. In contrast, in EBV-positive lymphomas and lymphoproliferative diseases, the impact of viral infection on the TME is not so evident and viral infection was rather assigned a predominantly oncogenic role [1,139,140,141,142]. Since the effects of EBV on the TME of EBV-associated malignancies is broad, this review focuses on the differences in the composition of the cellular and soluble components of the TME in both EBV-positive and EBV-negative malignancies.

#### 4.3.1. Cellular Composition of the TME

The immune cell composition of the TME, which is modulated by the expression of interleukins (ILs) and chemokines, partially overlaps between EBV-positive and EBV-negative malignancies [143,144]. Despite the high diversity, some features could be even linked to EBV-positive lymphoproliferations and lymphomas. By comparing the TME of different lymphoma entities, such as EBV-positive BL, DLBCL and cHL, a few similarities of the immune cell repertoire exist, which are not only based on the histomorphology. A main feature of cHL is the predominance of bystander immune cells with only a sparse presence of neoplastic Hodgkin–Reed/Sternberg cells, and thus it presents a unique pattern of a surrounding immune ecosystem [145]. In contrast, the malignant cells in BL and DLBCL represent the largest proportion within the tissue, and the non-neoplastic immune cells represent the minority of cells [146,147]. Despite these features, common characteristics between EBV-associated malignancies exist including high levels of tumor-infiltrating lymphocytes (TILs) within the TME [23,114,126,127,128,129,130]. Not only the amount of TILs is higher in EBV-positive tumors, but also the proportion of the different immune cell subpopulations within the TME varies. For example, the number of CD8^+^ T cells and M2-polarized tumor-associated macrophages (TAMs) is increased in both EBV-positive carcinoma and lymphatic malignancies including cHL, BL and DLBCL [23,130,131,145,148,149]. Higher numbers of CD8^+^ T cells are associated with an increased frequency of effector T cells expressing the cytotoxic molecules TIA and granzyme (gran) B and a reduced expression of the WNT and TGF-β pathway signature [114]. Recently, single-cell sequence analysis of CD8^+^ T cells from both the TME and the peripheral blood of EBV-positive NPC identified high numbers of exhausted CD8^+^ T cells, which in turn also contribute to a reduced cytotoxic activity [150,151]. In different EBV-positive tumor types, such as NPC [150,151], BL [145], DLBCL [152,153] and cHL [154], a significantly more restricted T-cell receptor (TCR) repertoire was found when compared to that of EBV-negative malignancies. T-cell exhaustion represents one of the most prominent strategies of tumors to circumvent the anti-tumor immune responses, but the underlying mechanisms of this phenomenon remain largely unknown [155].

Furthermore, the frequency of γδ T cells in patients with EBV-positive NPC was unaltered but showed an impaired T-cell function characterized by reduced cytotoxicity for the NPC targets [156]. In addition, a higher frequency of Tregs was shown in some EBV-associated malignancies [157,158,159,160], while Tregs are only poorly represented in BL [145]. Tregs are potent suppressors of other immune cells and can create an immunosuppressive environment [160]. In some EBV-positive malignancies, lower frequencies of CD8^+^ T cells as well as of other effector immune cells, such as NK cells and M1 macrophages, were found [158,160].

Concerning TAMs, increased numbers of CD163^+^ M2 TAMs [130] known to be involved in tumor progression and immune-suppressive functions were detected in almost all EBV-associated malignancies [130,145,149] and were mainly distributed in the stroma. In addition, myeloid-derived suppressor cells (MDSCs) with immune regulatory properties are expanded in EBV-positive tumors, such as NPCs [161], and have a potent immune-suppressive activity sustaining an anti-inflammatory TME by suppressing T-cell effector functions [150,162]. Next to their presence in the stroma, circulating MDSCs were detected in the PBMCs of HL, NPC and GC patients [163,164,165]. Cancer-associated fibroblasts (CAFs) generally surround tumor cells, in particular NPC, but with a variable extent and were also found in EBV-positive GC [166,167]. Upon stimulation, CAFs produce pro-inflammatory cytokines thereby creating a tumor-supportive micromilieu. Furthermore, the interplay of the suppressive immune cell subpopulations of the TME of EBV-positive tumors can strongly suppress the activation of effector T cells. This is also attributed to the effects of the EBV-encoded LMP-1 protein that is expressed in many tumor cells of EBV latency type II and III [148]. Despite that NK cells play a major role in anti-tumor immune responses, there exists only limited information in EBV-associated malignancies. An increased frequency of NK cells was found in the TME of EBV-positive HL [157] in association with the induction of tumor-suppressive pathways [168,169]. Furthermore, NPCs are highly infiltrated by NK cells with impaired function mitigated by elevated IL-18 levels, which was correlated with a worse patient prognosis [170,171]. In addition, the frequency of CD57^+^ NK cells is higher in EBV-infected than in EBV-negative GC, related to patients’ age, tumor diameter and PD-L1 expression [172] and linked to high tumoral IFN-γ levels [173]. NK cells are part of the innate immune response against EBV and have an important function during primary infection [174]. 

So far, the role of B cells in the TME of EBV-positive malignancies has been postulated by some authors, but this is controversially discussed. In the majority of EBV-associated malignancies, B-cell infiltration was found, but with a lower frequency when compared to T cells. In NPC, B-cell frequency highly varies with lower B-cell frequencies in one report [151] but higher frequencies in another study [150]. So far, there exist no investigations focusing on B-cell subpopulations including differences regarding EBV-positive or EBV-negative reactive B cells within the TME and their influence on the composition and functional condition. It is noteworthy that the altered immune cell composition in EBV-associated diseases is accompanied by a distinct patient outcome that is discussed in Section 4.5. The immune cell subpopulations, their differential regulation within the TME, the expression of immune-modulatory molecules and the clinical significance are summarized in Table 4.

#### 4.3.2. Soluble Mediators

An aberrant chemokine and cytokine expression directly affects the above-mentioned composition of the TME, and these soluble mediators have been shown to be involved in EBV-mediated malignancies. Relevant ILs and chemokines, their regulation and function in the tumor progression and local immune regulation of EBV-positive tumors are summarized in Table 5. In particular, the local secretion of pro-inflammatory ILs and chemokines influences the functionality of the immune cells and thus modulates tumor progression (Table 5) [183,184]. It can be assumed that alterations in these soluble mediators precede immune cell infiltration.

Studies of both solid and hematological EBV-associated malignancies demonstrated higher levels of IL-10 expression compared to EBV-negative malignancies in the same anatomical region or the same cells of origin [200,201,202,203,204], which are linked to local immune suppression or local enhanced immune cell tolerance [205]. In fact, IL-10 is known to (i) downregulate the expression of major histocompatibility complex (MHC) class I and II antigens of antigen-presenting cells (APCs) [191], (ii) induce Tregs [190], which in turn (iii) inhibit T-cell proliferation and IFN-γ secretion [192], and (iv) functionally block cytotoxicity of CD8^+^ T cells [188] causing an immune-suppressive TME. Furthermore, in vitro knockdown of IL-10 switches latent EBV-infected tumor cells to the lytic form leading to tumor cell death. This also synergizes with chemotherapy and thus leads to cell death [206] suggesting that IL-10 exhibits a key role in EBV-positive malignancies. Furthermore, IL-4, IL-6 and IL-13 are frequently upregulated in EBV-positive malignancies and might have relevance in the development and/or maintenance of EBV-associated malignancies [188,207,208]. Regarding IL-1β, increased levels were found in the TILs of EBV-positive GC [185]. Higher levels of IL-6 are present in both epithelial and lymphoid EBV-positive neoplasms acting as a growth factor [186,187,188]. In addition, IL-6 promotes Th17 differentiation, which is linked to autoimmunity, while their role in the context of malignancies is still controversially discussed [209]. Notably, tumor tissues frequently show a considerable infiltration of CD4^+^ T cells, which may act as a source of these cytokines [158]. Moreover, the IFN-γ-induced protein 10 (IP-10, CXCL10) is upregulated in several EBV-positive tumors, such as cHL, NPC and GC. The expression of IP-10 in the TME can attract a variety of immune cells, e.g., monocytes, T cells and NK cells [195,210]. In EBV-positive GC, IP-10 is linked to better survival [194,196,199]. Another frequently upregulated soluble mediator is the stromal cell-derived factor-1 (SDF-1, CXCL12) known to initiate chemotaxis and promote cell growth [197,198,199]. CAFs have been shown to secrete high levels of SDF-1/CXCL12 in different cancer types including EBV-positive NPCs [198,211,212,213]. In line with the increased expression of SDF-1/CXCL12, an upregulated expression of the associated chemokine receptor CXCR4 was also detected on EBV-positive NPC, GC and B-cell lymphomas [197,214,215].

#### 4.3.3. Extracellular Vesicles in EBV Infection and Persistence

As outlined above, EVs/exosomes released from EBV-infected cells have been shown to contain a variety of bioactive molecules [216] and may affect the phenotype of recipient cells depending on the cargo transmitted, thereby influencing the tumor progression [217]. In general, exosomes are able to stimulate immune responses by acting as antigen-presenting vesicles [218,219]. EBV-positive tumor cells can release EBV-encoded RNAs, EBER1, EBER2, miRNAs and the LMP-1 protein via exosomes allowing the RNAs and/or proteins to be captured by EBV-negative bystander cells, such as follicular DCs, and can then be presented by these APCs to other EBV-negative immune cells thereby inhibiting their activation [220]. In addition, exosomes of EBV-positive cancer cells can modulate the function of stromal cells, inhibit DC maturation in vivo and exhibit T-cell inhibitory activity [179,221,222] (see also Figure 2).

### 4.4. Immune Escape of EBV-Infected Cells

It has been well established that cells of the innate and adaptive immune system are able to recognize and eliminate cancer cells. However, the processes of malignant and/or viral transformation, as well as increased growth, are associated with strategies of tumor cells to escape immune surveillance [223] by evading the immune surveillance and/or by suppressing the hosts’ immune system mediated by shaping the TME and dampening the effector function [224,225]. Indeed, tumor cells can convert immune cells to a tolerogenic and dysfunctional state due to cell–cell interactions, soluble mediators and physical factors in the TME. In order to allow the establishment of infection and persistence, EBV has adopted different strategies to modulate signaling pathways to minimize their anti-viral activity while taking the advantage of their growth-promoting effects to circumvent host immune responses and to compromise innate and adaptive immunity during the latent and replicative phase of its life cycle [226]. The immune escape mechanisms are broad and include prevention of apoptosis induction, EBV-enhanced cell proliferation and inhibition of immune recognition of EBV-infected cells by latent or lytic EBV-encoded gene products and/or non-coding viral RNAs. These factors contribute to the disease prognosis and prediction of therapy response thereby exerting a clinical relevance [126]. Interestingly, there exist similarities but also differences between the immune escape strategies of solid and hematopoietic EBV-associated tumors [143,224].

A number of EBV gene products and miRs have been identified during the last decades to be involved in immune escape mechanisms of EBV-associated malignancies. The functions of the key molecules in the immune escape are addressed in this section in more detail (Table 6).

#### 4.4.1. Distinct Genetic Alterations of Immune-Modulatory Molecules

Some structural and epigenetic peculiarities have been described in EBV-positive malignancies, which represent molecular heterogeneous diseases, and were associated with immune escape mechanisms. A higher burden of genomic alterations in genes involved in immune signaling pathways, such as the JAK/STAT pathway (e.g., chromosomal gains of JAK2) [227,228] or the antigen presentation machinery (APM), has been identified. The latter include inactivating mutations and structural variants in the MHC class I genes, APM components (β_2_-M, TAP1) and the transcription factor NLRC5 known as the regulator of the expression of APM components, which are very common in EBV-positive NPC and NK/T cell lymphomas [116,229]. In contrast, genetic alterations of HLA class II genes are not frequently detected.

Loss-of-function mutations in negative regulators of the NF-kB pathway including TRAF3, CYLD and NFKBIA with consecutive increased NF-kB activation were detected in particular in NPC [104,115]. Furthermore, a high frequency of PD-L1/PD-L2-involving genetic aberrations, such as PD-L1/PD-L2 amplifications at chromosome 9p24.1, was demonstrated in EBV-positive B- and T-cell lymphoma but also in NPC [230].

#### 4.4.2. Downregulation of HLA Class I and Class II Surface Antigens

Next to structural alterations of HLA class I antigens, a frequently observed mechanism enabling immune evasion is the downregulation of components involved in the antigen presentation mediated by HLA class I. Many gene products of the APM are known to be transcriptionally induced by cytokines, especially by IFN-α, IFN-γ and TNF-α, while they are reduced by TGF-β and IL-10 and/or post-transcriptionally regulated by miRs [231,232]. In the context of EBV-associated malignancies, the expression of HLA class I APM can be influenced by host as well as EBV gene products. Several EBV-encoded molecules have been identified to directly interfere and reduce proper antigen presentation of EBV-infected cells. These include EBV-encoded proteins and miRs. EBV-encoded gene products expressed in the early or late lytic phase of EBV infection, such as BGLF5, BNLF2A and BILF1, downregulate the HLA class I antigen presentation [233,234,235,236,237]. Recently, multiple conserved residues in the extracellular regions have been shown to be important for the BILF1-mediated downregulation of HLA class I antigens [238]. EBNA-1 disrupts the peptide generation and their transport for functional presentation on HLA class I molecules [239], while BNLF2 interferes with the functionality of the peptide transporter (TAP)1 subunit [240,241], which is required for the peptide transport into the ER for peptide loading onto HLA class I molecules. In the prelatent phase, EBNA2 and lytic gene products are expressed [242], which prevent accurate peptide generation, transport and loading onto HLA class I molecules and thus impair the presentation of viral antigens toward CD8^+^ T cells able to distinguish between self- and non-self-antigens [243]. Peptide presentation is also negatively regulated by virus-encoded miRs, in particular, EBV-miR-BART17 and the three EBV-miR-BHRF1-3 [244]. Not only HLA class I but also HLA class II antigens could be affected by EBV infection. Gp42 known as an entry receptor of EBV has immune-suppressive properties, since both membrane-bound and soluble Gp42 block HLA class II antigen presentation, while BGLF5 degrades HLA class II [245,246,247]. Furthermore, Zta1 expression caused downregulation of HLA class II expression via inhibition of CIITA [248].

Concerning host gene expression, an upregulation of host miRs targeting components involved in the HLA class I antigen presentation pathway, so-called immune-modulating miRs have been identified [231,232]. The induced expression of host and/or virus-encoded immune-suppressive anti-inflammatory cytokines, such as IL-10 or the viral interleukin-10 (vIL-10), which is encoded by the EBV BCRF1 gene [249,250], downregulates the peptide presentation resulting in an inhibition of immune effector cells [251].

#### 4.4.3. Upregulated Expression of Non-Classical HLA Class I Antigens

In addition to classical HLA class I molecules, the non-classical HLA-G, HLA-E and HLA-F antigens not only present peptides but also represent ligands to inhibitory receptors of immune effector cells and mediate, under physiological conditions, the immunological tolerance of immune-privileged tissues, such as cornea, testis and chorion [252]. The HLA-G binds to the inhibitory receptors ILT2, ILT4 and KIR2DL4 that are expressed on different immune cell subsets such as NK cells, T cells, B cells or monocytes thereby representing an important immune escape mechanism [253]. So far, little information exists regarding the role of non-classical antigens upon EBV-driven malignant transformation. Interestingly, EBV infection can induce HLA-G [254], which is accompanied by an induction of IL-10 known to enhance HLA-G expression [255]. Furthermore, HLA-G is expressed in cHL and has been suggested as a potential immune escape mechanism in this disease [256]. HLA-E binds to the inhibitory receptors CD94/NKG2A and CD94/NKG2B and to the activating receptor NKG2C [257,258]. EBV-derived peptides have been detected to be presented by HLA-E molecules and to stabilize their surface expression [259]. In addition, HLA-F can be induced upon EBV-mediated transformation of lymphoblastoid cell lines [260]. In contrast, EBV-encoded proteins or miRs altering the expression of HLA-G, -E and -F in EBV-associated malignancies have not yet been identified.

#### 4.4.4. Increased Expression of Checkpoint Molecules

Next to HLA antigens, immune-modulatory cell surface proteins also exhibit a distinct expression pattern in EBV-positive tumors. Regarding immune checkpoints, such as the programmed death-1 receptor (PD-1) and its programmed death ligand-1 (PD-L1) and PD-L2, which represent one prominent mechanism to escape immune surveillance, an increased PD-L1 expression was detected in EBV-positive when compared to EBV-negative specimens from NPC [139], gastric cancer [261], DLBCL and cHL [183,262], respectively. PD1 and CTLA-4 are upregulated on EBV-infected T cells, and PD-L1 expression is increased on EBV-infected lymphoma cells. Regarding the underlying molecular mechanisms, higher PD-L1 and/or PD-L2 expression levels were associated with a gain of chromosome 9p24 on neoplastic cHL cells [263,264], since PD-L1 is induced by the EBV-encoded LMP1 through the NF-ĸB pathway in NK/T cell lymphoma [265] and in DLBCLs by the EBNA2 gene product. The latter was confirmed by in vitro studies of EBNA2 overexpression leading to an upregulation of PD-L1 due to downregulation of the PD-L1 targeting human endogenous miR-34a [266]. However, it is noteworthy that the EBV miR-BHRF1-2-5p has a counter-regulatory role by fine-tuning the LMP1-mediated upregulation of PD-L1 and PD-L2 [267]. Furthermore, the expression levels of TIM-3, LAG-3 and VISTA were upregulated in EBV-specific T cells, which is accompanied by the impairment of LMP1/2-specific T-cell function, and were directly associated with a high viral load [268,269]. However, with the exception of the PD1/PD-L1 system, the expression levels and the role of immune checkpoints in EBV-associated malignancies have not yet been systematically evaluated.

#### 4.4.5. Altered Regulation of Inflammatory Signal Transduction Pathways and Soluble Factors

Inflammatory signaling processes play a key role in some EBV-associated malignancies, e.g., NPC tumorigenesis. These include the activation of the NF-ĸB signaling pathways, which affect growth properties but also immune evasion by modulating the composition of the TME. During EBV infection, various pattern recognition receptor signaling pathways are activated and targeted by latent and/or lytic EBV proteins or by EBV-specific miRs to minimize their anti-viral activity as summarized in Table 6. For example, BGLF5, which is expressed during the productive phase of infection, leads to the RNA degradation of immunologically relevant proteins, such as the Toll-like receptor (TLR) 2 and TLR-9 [245]. The TLR signaling pathways can activate NF-ĸB, which is controlled by various post-translational modifications [270]. These modifications could be affected by EBV-encoded proteins, such as BGLF4, which suppresses NF-ĸB activity and by BPLF1, which reverses the ubiquitination of TLR signaling intermediates [245]. Furthermore, LMP1 downregulates the expression of the tumor necrosis factor (TNF) α, [134,271]. Another pro-inflammatory cytokine signaling inhibited by the miR-BHRF1-2-5p targets the IL-1 receptor 1 thereby interfering with the pro-inflammatory signaling required for early activation of components of the innate immune system [272]. While IL-6 and IL-12 are regulated by 5 EBV-specific miRs in early infection leading to a suppressed Th1 differentiation [273], the chemokine CXCL11 is targeted by BHFR1 miRs thereby affecting the recruitment of CD8^+^ T cells [274]. Furthermore, the interferon (IFN) signaling pathway is altered by EBV infection. A number of EBV-encoded genes can affect different components of the IFN signal transduction pathways, such as STAT1, IFN receptors, IFN-stimulated genes (ISGs) and IFN-regulatory factors (IRFs), as summarized in Table 6.

**Table 6 cancers-13-05189-t006:** Immune escape mechanisms and EBV-encoded gene products involved.

Component	Gene Product	Mechanism	Reference
HLA class I	BGLF5	degradation of HLA class I	[235,245]
	BNLF2a	inhibition of peptide transport by blocking	[233,240,241]
	BILF1	impaired HLA class I export increased turnover of HLA class I surface molecules	[234,237,238,275]
	BCRF1 (vIL-10)	inhibition of HLA class I	[251]
	EBNA1	disruption of peptide generation and transport	[239]
	EBNA2	inhibition of peptide generation and presentation	[243]
HLA class II	gp42 (BZLF2)	block of TCR/HLA class II interaction	[246,247]
	BGLF5	degradation of HLA class II	[245]
	Zta	inhibition of CIITA promoter activity, posttranscriptional regulation by impairing function of the invariant chain	[248]
	BCRF1 (vIL-10)	inhibition of HLA class II expression	[276]
Checkpoints	LMP1	induction of soluble PD-L1	[277]
TLR/NF-ĸB pathway	BGLF5	downregulation of TLR-2 and -9	[245]
	BZLF1	inhibition of NF-ĸB	[278]
	BGLF4	suppression of NF-ĸB activity	[279]
	BPLF1	inhibition of NF-ĸB activation by reversion of ubiquitination of TLR signaling	[280,281]
	EBNA1	inhibition of NF-ĸB	[282]
	LMP1	reduction of TLR-9	[283]
	LMP2	inhibition of NF-ĸB	[284]
IRFs/IFN type I signaling	BZLF1	inhibition of IRF7 transcriptional activity; reduction of IFN-ƴ receptor	[271,285]
	BILF4	inhibition of IRF7	[286]
	BGLF4	inhibition of IRF3	[287]
	BRLF1	reduction of IRF3 and IRF7 expression	[288]
	EBNA1	modulation of STAT1 signaling	[289]
	EBNA-2	inhibition of ISGs	[290,291]
	LMP-1	regulation of STAT1	[292]
	LMP-2a	inhibition of JAK/STAT signaling, acceleration of IFN receptor turnover	[293]
	LMP-2b	acceleration of IFN receptor degradation	[293]

#### 4.4.6. Downregulated Molecules Modulating Innate Immune Cells

Innate immune cells play an important role in the host response against viral infections including EBV [294]. Immune responses directed against virus-associated malignancies are also modulated by downregulation of ligands for activation receptors of immune effector cells, in particular the NKG2D ligands including MICA/B as well as ULBP1-6. The expression of several ligands of the activation receptor NKG2D, which is induced upon, e.g., cellular stress, viral infections and inflammation, is downregulated upon EBV infection, such as MICA and MICB. This is controlled by EBV-miR-BART7 and EBV-miR-BART2-5p [135,295] negatively interfering with NK cell responses. The enhanced recruitment and activation of Tregs, the subsequent anergy of CD8^+^ cytotoxic T cells, the crosstalk with tumor-growth-promoting M2 macrophages and the overexpression of the immune-suppressive enzyme indoleamine 2,3-deoxygenase (IDO) are also involved in immune evasion [296,297,298].

### 4.5. Clinical Relevance of EBV in Malignancies

Based on the diverse immune escape strategies described, a link of EBV infection with disease progression and response to therapy including immunotherapies is obvious. EBV-positive NPCs have been associated with significantly worse survival compared to EBV-negative patients [299,300], which is due to the induction of an immune-suppressive TME. This is in line with many other different EBV-induced tumor types, in which the immune-suppressive TME has been correlated with the patient outcome. In contrast, higher numbers of TILs have been shown to be associated with an improved survival of EBV-associated NPC, GC and ICC patients [23,114,150]. Thus, there exists growing evidence that the prognosis of patients with EBV-associated malignancies is influenced by different immune cell populations. While increased numbers of effector T cells, B cells and NK cells have been related to better survival [119], higher numbers of Tregs, inhibitory DC subsets and M2-polarized TAMs known to support an immune-suppressive TME are associated with shorter disease-free survival [128,149,301]. Of note, BLs with a pro-inflammatory TME characterized by higher levels of M1-polarized TAMs and associated granulomatous reaction have a favorable prognosis and an occasionally spontaneous regression [302]. However, not only the frequencies of the different immune cell subsets but also their functional status are of clinical relevance. Exhausted T-cell subsets and increased numbers of highly dominant T-cell clones with a consecutive limited TCR repertoire are contributing to an immune cell dysregulation supporting the immune surveillance and have been shown to be associated with worse patient survival [152]. Since the expression of immune checkpoints is one key mechanism of tumors to escape the host immune system, it is not surprising that the expression of PD-L1 is associated with the patient prognosis. However, its prognostic significance remains controversial, and the impact on the prognosis is dependent on the tumor type [103,114,127,178]. In DLBCL and cHL, EBV-positive tumors with high PD-L1 expression showed a significantly shortened overall survival compared to other subtypes [303]. Furthermore, a high PD-L1 expression correlated with high EBV copy numbers suggesting that the EBV load contributes to the expression levels of checkpoint molecules [304]. Low-dose anti-PD1 antibodies were highly efficacious and safe in patients with relapsed/refractory cHL and improved their survival [305]. In EBV-positive NPC and GC, higher PD-L1 expression was associated with improved survival [300,306]. An enhanced efficacy of PD-L1/PD-1 blockade was shown in patients, which was accompanied by numbers of M1 TAMs [307] further suggesting EBV-positive NPC and EBV-positive GC as ideal candidates for PD-1-directed therapies [308]. Furthermore, the antigen presentation via HLA class I molecules is deregulated in many EBV-positive malignancies thereby affecting anti-tumoral immune responses, which has been also associated with worse patient survival [150].

Taken together, a strong impact of intrinsic and extrinsic immune escape mechanisms, in particular, on the local immune cell composition has been demonstrated. Therefore, different immune therapeutic approaches have been investigated in preclinical and clinical studies. Next to the immune checkpoint blockade, the spectrum of therapeutic strategies encompasses anti-viral drugs, small molecules, cytokines and cellular vaccination or T-cell therapy. Combination therapies of chemotherapy and anti-viral drugs, such as ganciclovir, resulted in a strong synergistic effect with higher cytotoxicity compared with chemotherapy alone and subsequently led to disease stabilization [309].

It has been demonstrated that EBNA-1 inhibitors are potent and selective inhibitors of cell growth in tissue culture and animal models of EBV-positive GC suggesting that pharmacological targeting of this latent EBV gene product may be an effective strategy to treat patients with EBV-positive GC [310]. Furthermore, a combination of betulinic acid and Chidamide (CDM, CS055), a novel histone deacetylase inhibitor (HDACi), could significantly inhibit EBV replication with ROS over-generation and subsequent DNA damage and apoptosis [311]. First studies with an LMP-2 DC vaccine in patients with NPC showed specific CD8^+^ T-cell responses directed against LMP-2, which plays a critical role in controlling and preventing the recurrence and metastasis formation of NPC [312]. Currently, there are multiple trials testing the use of both donor-derived and third-party EBV- specific T cells in the setting of treatment-refractory EBV-positive malignancies. First trials with chimeric antigen receptor (CAR) T cells against latent EBV antigens but also against early lytic viral gene products have been carried out. These CARs appear to be protective for the control of EBV infection and EBV-mediated oncogenesis thereby providing a novel promising therapeutic strategy against EBV-associated malignancies [53,313,314].

## 5. Conclusions

The Epstein–Barr virus was first described in 1964, in African endemic BL samples [1]. However, more than 50 years later, the detailed mechanisms of EBV in disease initiation and progression are still not completely understood. However, there exists strong evidence that EBV latent gene and miR expression simultaneously target different intracellular pathways in EBV-infected cells and thereby modulate the TME to the gene’s benefits. After an unusual immune-cell-rich TME was initially observed, different immune cell subpopulations and their function were examined, and a predominantly immunosuppressive TME could be proven in the following years. Interestingly, a highly variable but distinct composition of the TME with increased numbers of effector T cells and Tregs was observed [23,131,148]. This is also supported by soluble factors such as IL-10 which is upregulated in most EBV-positive tumors thereby inducing Tregs, which results in suppression of effector T-cell function [185,188,190,191,192]. In addition, the EBV latency gene product LMP-1 suppresses the function of effector T cells and can be presented by tumor cells, extracellular vesicles or antigen-presenting DCs. Other strategies of the tumor cells to escape the immune surveillance are the downregulation of MHC class I molecules or upregulation of non-classical HLA-G. Furthermore, higher expression levels of the immune checkpoint molecules PD-L1 have been shown in almost all EBV-positive tumors and have been shown to be regulated by EBV. It is obvious that EBV-positive tumors are able to develop a multitude of mechanisms to escape immune surveillance. Thus, one single therapeutic strategy might be often not sufficient for disease control. Increased insights into the TME and tumor-intrinsic immune escape strategies will help to design and improve (immuno) therapeutic strategies. Currently, immune checkpoint blockade, such as PD-1/PD-L1 antibodies, small molecules targeting EBV latency gene product, cellular vaccination and CAR T-cell therapy against EBV antigens seem to offer promising therapy options [53,308,310,311,312].

## Figures and Tables

**Figure 1 cancers-13-05189-f001:**
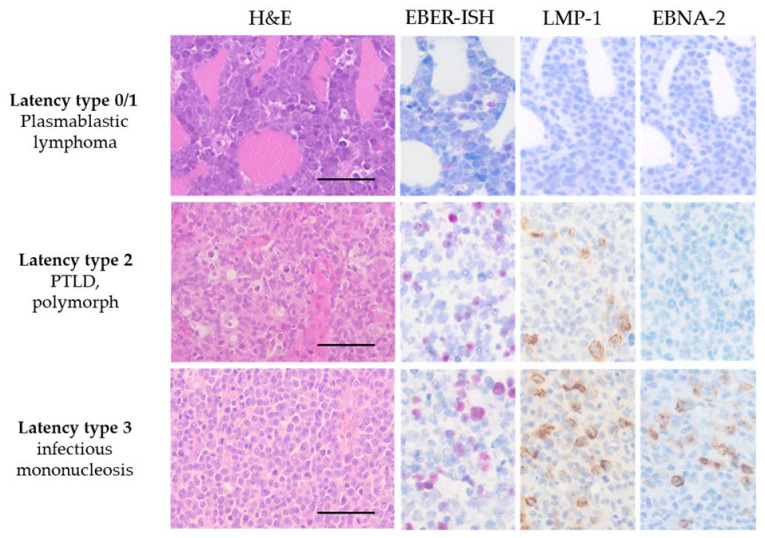
Micrographs of EBV-associated diseases using histopathological techniques. EBV latency type is correlated with EBV-associated diseases by using EBER-ISH and immunohistochemical stainings with antibodies against LMP-1 and EBNA-2. All latency types show positive signals in the EBER-ISH (pink nuclear signals). Plasmablastic lymphoma, latency type 0 or 1, is negative for LMP-1 or EBNA-2. In contrast, the polymorphic PTLD exhibits positive signals for LMP-1 (brown membranous signal), while immunohistochemistry (IHC) for EBNA-2 remains negative. In infectious mononucleosis, all three markers are positive, and thus a latency type III is determined. H&E stained micrographs show scale bars representing 50 µm.

**Figure 2 cancers-13-05189-f002:**
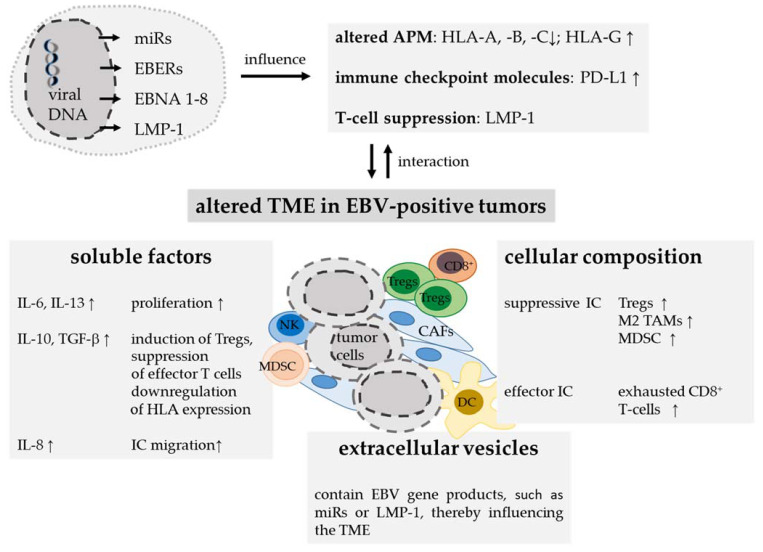
Composition of the TME in EBV-positive tumors. EBV gene products influence innate and adaptive immunity and thereby modulate the TME. Latent membrane protein (LMP) 1 suppresses the function of effector T cells and can be presented by tumor cells, extracellular vesicles or antigen-presenting cells (APCs) such as dendritic cells (DCs) via human leukocyte antigen (HLA) class II molecules. In addition, EBV-genome-related miRs influence the HLA class I antigen processing machinery (APM). Other strategies of the tumor cells to escape the immune surveillance are upregulation of non-classical HLA-G and/or immune checkpoint molecules, such as the programmed death ligand-1 (PD-L1). Furthermore, EBV-positive tumors induce a highly variable composition of the TME with increased numbers of different immune cell subsets, in particular high frequencies of effector T cells, regulatory T cells (Treg) and M2-polarized macrophages. It has been shown that soluble factors, secreted by tumor cells, cancer-associated fibroblasts (CAFs) or even other immune cells (ICs) promote immune cell migration into the TME. Furthermore, soluble factors such as interleukin-10 (IL-10) can induce Tregs, suppress effector T cells and regulate HLA class I expression.

**Table 1 cancers-13-05189-t001:** Epidemiological features of EBV-associated neoplasia.

Disease	Prevalence of EBV^+^ Diseases	Frequency of EBV Infection within the Tumor Entity	Age at Disease Onset	Geographic Distribution	References
Carcinomas					
nasopharyngeal carcinoma (NPC)	1:100,000 in Europe 25:100,000 in Asia	~90%	18–80 years	South East Asia, North and East Africa	[17,22]
gastric adenocarcinoma (GC)	1:100,000	~9%	frequently < 60 years	worldwide	[18,27,28]
intrahepatic cholangiocarcinoma (ICC)	<0.01:100,000	6.6%	24–68 years	South East Asia	[23,29,30]
Lymphomas and lymphoproliferative disorders					
chronic active EBV infection (CAEBV)	rare	100%	5–31	Asia	[31,32]
mucocutaneous ulcer	unknown	100%	>60 years	worldwide	[25,33]
lymphomatoid granulomatosis	rare	~100%	typically adults, rare in childhood	Europe and North America	[34,35]
extranodal NK/T cell lymphoma (ENKTL)	rare	100%	17–89 years	Asia	[36,37]
classical Hodgkin lymphoma (cHL)	0.5:100,000 in Asia 2.3:100,000 in Europe	50–90% depending on the subtype	20–65	worldwide	[19]
plasmablastic lymphoma	0.1:100,000	80%	7–65 years	worldwide	[25,38]
post-transplant lymphoproliferative disease (PTLD)	<1% in bone marrow transplanted up to 30% in small bowel transplanted patients	60–80%	children more often affected	worldwide	[39]
angioimmunoblastic T-cell lymphoma (AITL)	0.05–0.2:100,000	~70%	20–86 years	Europe	[40,41,42]
primary effusion lymphoma	rare	70%	young adults	worldwide	[25]
Burkitt lymphoma (BL) sporadic	0.15:100,000	<15%	15–40 years	worldwide	[43]
Burkitt lymphoma (BL) endemic	3–6:100,000	>90%	2–20 years	Central Africa East Africa	[20]
diffuse large B-cell lymphoma (DLBCL)	5–7:100,000	Europe ~ 4% Asia ~ 15%	50–91 years	Asia	[24,25,44]
Soft-tissue tumors					
leiomyosarcoma associated with immune suppression	rare	100%	children and adolescents	worldwide	[45]
EBV^+^ inflammatory follicular dendritic cell sarcoma	rare	Unknown	8–77 years *	worldwide	[21,46]

* Data from 9 cases.

**Table 2 cancers-13-05189-t002:** EBV-associated gene and protein expression profiles, allocated latency type and related diseases.

Latency Type	Gene Expression Profile Associated with Latent EBV Expression	Diseases with Strong Association to a Certain Latency Type	Diseases with Variable Latency Types
0/I	EBERs EBNA-1, BART (miRs)	endemic or sporadic Burkitt lymphoma (BL), plasmablastic lymphoma, primary effusion lymphoma	nasopharyngeal carcinoma (NPC), astric adenocarcinoma (GC), intrahepatic cholangiocarcinoma (ICC),
			NK cell leukemia,
II	EBERs EBNA-1 LMP-1, -2A, -2B BART (miRs)	classical Hodgkin lymphoma, EBV-positive diffuse large B-cell lymphoma (DLBCL), not otherwise specified (NOS) extranodal NK/T cell	angioimmunoblastic T-cell lymphoma (AITL), chronic active EBV infection of T- and NK-cell type (CAEBV)
		lymphoma, leiomyosarcoma associated with	
		immune suppression	DLBCL associated with chronic inflammation
			mucocutaneous ulcer
III	EBERs EBNA-1, -2, 3A, -3B, -3C LMP-1, -2A, -2B BHRF1 BART (miRs)	infectious mononucleosis	lymphomatoid granulomatosis, post-transplant lymphoproliferative disorders (PTLD)

**Table 4 cancers-13-05189-t004:** Differences of the frequency of immune cell subpopulations in the TME in EBV-positive vs. EBV-negative malignancies and their clinical significance. Abbreviations: DC: dendritic cell; n.a. not available; NK cell, natural killer cells; TAM, tumor-associated macrophage; TIL, tumor-infiltrating lymphocyte; Treg, regulatory T cell.

Disease	Immune Populations/Markers	EBV+ vs. EBV−	Clinical Significance	Reference
NPC	TILs	higher	good prognosis	[114,128]
	CD8^+^ T cells	higher	good prognosis	[114,150]
	exhausted T cells	more frequent	n.a.	[151,175]
	Tregs	higher	n.a.	[150,160]
	B cells	higher	better prognosis	[114,150]
	LAMP3^+^ DC	higher	n.a.	[150]
	M2 TAM	high	poor survival	[149]
GC	TILs	higher	increased survival	[127,176]
	CD3^+^ T cells	higher	increased survival	[127,177]
	CD8^+^ T cells	higher	increased survival	[177,178]
	DCs	higher	correlation of some DC subsets with a worse survival	[179]
ICC	CD8^+^ T cells	higher	good prognosis	[23]
	CD20 B cells	higher	good prognosis	[23]
cHL	CD56^+^ CD16^+^ NK cells	higher	n.a.	[157]
	FoxP3^+^ Tregs	increased	worse prognosis	[180]
	exhausted T cells	more frequent	n.a.	[154]
	M2 TAM	higher	worse survival	[181]
PTLD	CD8^+^ T cells	high	none	[158]
	Tregs	high	n.a.	[158]
	TAM	high		[182]
DLBCL	TCR repertoire	increased highly dominant clones	worse survival	[152,153]
	M2 TAM	high	worse survival	[130]
BL	CD8^+^ T cells	higher	n.a.	[145]
	exhausted T cells	more frequent	n.a.	[145]
	M2 TAM	high	n.a.	[145]

**Table 5 cancers-13-05189-t005:** Altered cytokine and chemokine expression pattern, their functional relevance and detection methods in EBV-associated malignancies.

Interleukin	Origin	Regulation	Function	Detection Methods	Reference
IL-1ß	predominantly secreted by monocytes and macrophages	upregulation	inflammation	IHC, FFPE tissues	[185]
IL-4	predominantly secreted by Th2 cells, mast cells, NKT cells, basophils and eosinophils	upregulation	cell growth	RNA, PBMNCs	[38]
IL-6	predominantly secreted by monocytes, macrophages, T/B cells, neutrophils, endothelial cells, fibroblasts, adipocytes	upregulation	cell growth, Th17 differentiation	IHC FFPE tissue, WB; ELISA: cell culture	[186,187,188]
IL-8	predominantly secreted by monocytes and macrophages	upregulation	cell migration	IHC, FFPE tissue	[189]
IL-10	predominantly secreted by regulatory T cells, macrophages, DCs and neutrophils but also Th2 cells and Th17 cells	upregulation	immunosuppression, downregulation of MHC class I, induction of Treg	ELISA: PMBC cell culture	[185,188,190,191,192]
IL13	predominantly secreted by Th2 cells, mast cells, basophils	upregulation	cell growth	ELISA: cell culture	[193]
IFN-γ	predominantly secreted by NK cells, CTLs, Th1 cells	upregulation	inflammation	IHC: FFPE tissue	[185]
IP-10 (CXCL10)	predominantly secreted by monocytes and macrophages but also by endothelial cells	upregulation	inflammation, chemotaxis	RNA: cell culture, tumor tissue; ISH: FFPE tissue	[194,195,196]
SDF-1, CXCL12	predominantly secreted by macrophages and adipose tissue but also by cancer-associated fibroblasts	upregulation	inflammation, chemotaxis, cell growth	IHC: FFPE tissue; flow cytometry: PBMC	[197,198,199]

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
