# Peer review of "Epstein–Barr Virus—Associated Malignancies and Immune Escape: The Role of the Tumor Microenvironment and Tumor Cell Evasion Strategies"

_cancers, 2021, doi:10.3390/cancers13205189_

Round 1

Reviewer 1 Report

The authors have adapted the paper adequately.

Author Response

Dear Reviewer,

thank you very much for your final decision.

best regards

Marcus Bauer and Barbara Seliger

Reviewer 2 Report

Overall, The Manuscript has improved significantly. It is generally easier to read, especially the viral life cycle, which was previously very unstructured. In addition, the authors have fully addressed my main criticisms. I have only one minor point: in lines 99-102, there is still some confusion: GC is repeated twice, first associated with 18-80 years and then with <60 years in the same sentence.

Author Response

Dear Reviewer,

thank you very much for your kind comments. We changed the above mentioned sentence the following:

In contrast, EBV-positive nasopharyngeal carcinoma (NPC) and gastric carcinoma (GC) occur in adults between 18 - 80 years with EBV-positive GC patients frequently younger than 60 years.    We hope that the changes will make the statement easier to understand.   Best regards   Marcus Bauer und Barbara Seliger

Reviewer 3 Report

This manuscript by Bauer and others is a revised version of a review article. Although it seems that the manuscript has improved to some extent after revision, I still have to say that there are so many inaccurate or inappropriate citations and expressions. This time, I reached page 10, but this is it. Too many inappropriate citations, which means scientific uncertainty, and I lost my motivation now. The authors must check if the citations are good or not by themselves, one by one.

Page 3 line 75: ref4 (Smatti et al) deals only with LMP1, and thus must be removed here. Ref 6 (Liebowitz et al) deals only with LMP genes, and again must be removed.

Page 3 line 78: ref 7 (Rivailler et al) is inappropriate because it is about a virus of new world monkey, not EBV.

Page 3 line 87: ref14 (Abbott et al) must be replaced with appropriate one.

Page 3 line 89: ref15 (Munz et al) deals with non-coding RNAs and must be replaced with more appropriate one.

Page 3 line 100, 101: “while” appears three times in one sentence, which should not be the best way.

Page 4 line 124: “non-transcribed” is odd. Does the author mean “non-coding”?

Page 5 line 143: ref 62 (Iizasa et al), 63 (Seto et al) are about microRNAs and do not explain about the BCL2 homologs. Must be replaced.

Page 5 Table 2: right column, “(GC)” must be associated with gastric “gastric adenocarcinoma”.

Tables: Unifying of nomenclature is needed, overall. For example, Table 1 says gastric carcinoma and intrahepatic cholangiocellular carcinoma, while Tabel 2 mentions gastric adenocarcinoma and intrahepatic cholangiocarcinoma.

Page 6 Table 2: “EBNA-1 to 6” is confusing because they used other names in page 3 line 81 (EBNA-3A, -3B, 3C). Unifying of nomenclature is needed.

Page 6 Table 2: “angioimmunoblastic T cell lymphoma” should be “(AITL)” but not “(CAEBV)”.

Page 6 Table 2: what does it mean by the empty row, between “NK cell leukemia” and “angioimmunoblastic T cell lymphoma”. Also, btween “mucocutaneous ulcera” and “lymphomatoid granulomatosis”.

Page 6 line 191: abbreviation of “viral capsid antigen” must be “VCA”. The gp220 is an envelope glycoprotein, but not a capsid protein at all.

Page 6 line 196: what does it mean by this “it”? If it means “viral load in peripheral blood”, it must diminish sooner. This sentence must be re-written just so there is no confusion.

Fig. 1: polymorphoic PTLD is shown as a latency type 2 example, but Table 2 shows that PTLD exhibits latency III pattern. If polymorphoic PTLD is different from conventional PTLD, polymorphoic PTLD must also be included in the Table 2.

Page 8 line 273: ref 98 only cites ref 99, but G2 impairment by LMP1 is not the topic of this report. So ref 98 (Tsang et al) must be deleted.

Page 8 line 278: ref 102 (Tsimbouri et al) does not mention anything about DNA damage nor ROS. Must be removed.

Page 10 line 354: ref 133 (Seliger et al) and ref 134 (Lechien et al) must be replaced with appropriate ones because these reports deal with HPV but not EBV.

Page 10 line 357: it seems these citations (refs 24, 115, 127-131, 135, 136) do not support the sentence.

Page 10 line 363: ref 141 (Tsang et al) and ref 142 (Pathmanathan et al) do not cover lymphomas and lymphopriliferative diseases. Must be replaced with appropriate ones.

Author Response

Dear Reviewer,

thank you again for your critical comments. We changed some of the references as suggested. However, we do not fully understand the rejection of some of the references we used. We hope the changes are suitable.

Page 3 line 75: ref4 (Smatti et al) deals only with LMP1, and thus must be removed here. Ref 6 (Liebowitz et al) deals only with LMP genes, and again must be removed.

As told last time, we searched for better references and came to the conclusion that the quotation used is applicable and also in other works, e.g. IARC Monographs on the Evaluation of Carcinogenic Risks to Humans, No. 70, Lyon (FR): International Agency for Research on Cancer; 1997. was used

Page 3 line 78: ref 7 (Rivailler et al) is inappropriate because it is about a virus of new world monkey, not EBV.

You are right, the ref is not primarily about EBV, but they wrote a lot about EBV and point out that the exact function of some genes is not known.

Page 3 line 87: ref14 (Abbott et al) must be replaced with appropriate one.

Shannon-Lowe, C.; Rickinson, A. The Global Landscape of EBV-Associated Tumors. Front. Oncol. 2019, 9, 713, doi:10.3389/fonc.2019.00713.

Page 3 line 89: ref15 (Munz et al) deals with non-coding RNAs and must be replaced with more appropriate one.

IARC Working Group on the Evaluation of Carcinogenic Risks to Humans. Biological Agents. Lyon (FR): International Agency for Research on Cancer; 2012. (IARC Monographs on the Evaluation of Carcinogenic Risks to Humans, No. 100B.) EPSTEIN-BARR VIRUS.

Page 3 line 100, 101: “while” appears three times in one sentence, which should not be the best way.

We changed the sentence the following: In contrast, EBV-positive nasopharyngeal carcinoma (NPC) and gastric carcinoma (GC) occur in adults between 18 - 80 years with EBV-positive GC patients frequently younger than 60 years

Page 4 line 124: “non-transcribed” is odd. Does the author mean “non-coding”?

We changed the it to: EBV-encoded small RNAs (EBERs) and non-coding BART RNAs

Page 5 line 143: ref 62 (Iizasa et al), 63 (Seto et al) are about microRNAs and do not explain about the BCL2 homologs. Must be replaced.

You are right, ref. 62 and 63 deal with miRs, while in addition in ref. 63 the link to bcl-2 homologues is explained.

Page 5 Table 2: right column, “(GC)” must be associated with gastric “gastric adenocarcinoma”.

We changed it as suggested.

Tables: Unifying of nomenclature is needed, overall. For example, Table 1 says gastric carcinoma and intrahepatic cholangiocellular carcinoma, while Tabel 2 mentions gastric adenocarcinoma and intrahepatic cholangiocarcinoma.

We changed the nomenclature to gastric adenocarcinoma (GC) and intrahepatic cholangiocarcinoma (ICC).

Page 6 Table 2: “EBNA-1 to 6” is confusing because they used other names in page 3 line 81 (EBNA-3A, -3B, 3C). Unifying of nomenclature is needed.

We corrected it to: EBNA-1, -2, 3A,- 3B, -3C.

Page 6 Table 2: “angioimmunoblastic T cell lymphoma” should be “(AITL)” but not “(CAEBV)”.

We corrected the table, because there was something missing: angioimmunoblastic T cell lymphoma (AITL), chronic active EBV infection of T and NK cell type (CAEBV)

Page 6 Table 2: what does it mean by the empty row, between “NK cell leukemia” and “angioimmunoblastic T cell lymphoma”. Also, between “mucocutaneous ulcera” and “lymphomatoid granulomatosis”.

The table has been edited by the journal and I can´t fix the problem with the empty rows. The table initially submitted was as follows:

0/I

EBERs

Burkitt lymphoma (BL)

EBNA-1,

BART (miRs)

plasmablastic lymphoma

primary effusion lymphoma

nasopharyngeal carcinoma (NPC)

gastric adenocarcinoma (GC)

intrahepatic cholangiocarcinoma (ICC)

NK cell leukemia

II

EBERs

EBNA-1

LMP-1, -2A. -2B

BART (miRs)

classical Hodgkin lymphoma,

EBV–positive diffuse large B cell lymphoma (DLBCL), not otherwise specified (NOS)

extranodal NK/T cell

angioimmunoblastic T cell lymphoma (AITL)

chronic active EBV infection  of T- and NK-cell type (CAEBV)

lymphoma

leiomyosarcoma associated with

immune suppression

DLBCL associated with chronic inflammation

mucocutaneous ulcera

III

EBERs

EBNA-1 t EBNA-1, -2, 3A,- 3B, -3C  

LMP-1, - 2A, -2B

BHRF1

BART (miRs)

infectious mononucleosis

lymphomatoid granulomatosis

post-transplant lymphoproliferative dis-orders  (PTLD)

Page 6 line 191: abbreviation of “viral capsid antigen” must be “VCA”. The gp220 is an envelope glycoprotein, but not a capsid protein at all.

We changed the sentence as follows: In detail, the quantification of episomal EBV DNA coding for EBNA-1 and the viral envelop glycoprotein (gp)220 in the plasma and peripheral blood mononuclear cells (PBMCs) had to stand the test as a suitable marker of acute EBV infection that correlates with clinical symptoms [77,78].

Page 6 line 196: what does it mean by this “it”? If it means “viral load in peripheral blood”, it must diminish sooner. This sentence must be re-written just so there is no confusion.

We restructured the sentence the following: It has to be taken into account that the viral load is higher in the oral cavity than in peripheral blood [80].

Fig. 1: polymorphoic PTLD is shown as a latency type 2 example, but Table 2 shows that PTLD exhibits latency III pattern. If polymorphoic PTLD is different from conventional PTLD, polymorphoic PTLD must also be included in the Table 2.

PTLD can have different latency types (latency II or III patterns), which is why it is located in the "Diseases with variable latency types" column in Table 2.

Page 8 line 273: ref 98 only cites ref 99, but G2 impairment by LMP1 is not the topic of this report. So ref 98 (Tsang et al) must be deleted.

We deleted ref. 98.

Page 8 line 278: ref 102 (Tsimbouri et al) does not mention anything about DNA damage nor ROS. Must be removed.

Ref 102 refers to the sentence before, we changed the position.

Page 10 line 354: ref 133 (Seliger et al) and ref 134 (Lechien et al) must be replaced with appropriate ones because these reports deal with HPV but not EBV.

We changed the sentence the following: Although highly variable, the density of lymphocytes and plasma cells within the tumor stroma and of EBV-associated malignancies is elevated when compared to EBV-negative neoplasia as seen in other virus-associated tumors.

Page 10 line 357: it seems these citations (refs 24, 115, 127-131, 135, 136) do not support the sentence.

We changed the sentence the following: However, some characteristics are more common between EBV-associated tumors compared to EBV-negative counterparts, which do not depend on the anatomical localization or cellular origin

Page 10 line 363: ref 141 (Tsang et al) and ref 142 (Pathmanathan et al) do not cover lymphomas and lymphoproliferative diseases. Must be replaced with appropriate ones.

Küppers, R. B cells under influence: transformation of B cells by Epstein–Barr virus. Nat Rev Immunol 3, 801–812 (2003). https://doi.org/10.1038/nri1201

Best regards

Marcus Bauer und Barbara Seliger

This manuscript is a resubmission of an earlier submission. The following is a list of the peer review reports and author responses from that submission.

Round 1

Reviewer 1 Report

This review article by Bauer and others summarizes the relationships of EBV and immune system in malignancies. As I started reading the manuscript, I soon realized that there are too many inappropriate citations. Also, because there is no EBV or virology expert in the authors, and thus they seem to have many misunderstandings on EBV virology. Honestly, the current manuscript is an exhibition of incorrect or doubtful pieces of information, and I stopped reading it at page 4. I am afraid but this manuscript must be rejected once, and see if the authors could improve the manuscript. I may read page 5 and later of the manuscript only when the authors could cope with these minimum essential issues.

Examples of inappropriate references: ref 3, 4, 5, 6, 9, 12, 13, 14, 54, 58, 64, 65, 69. 

 page 2 line 66: "different virions are produced and can be distinguuished". What does it mean?

page 3 line 102-104:Wrong. No one has proved that circularized episomal DNA entered the nucleus. 

page 3 line 107-108: germline transmission of integrated EBV genome?? Highly unlikely. By the way, HHV-6 is not less common at all.

page 4 line 127: Is EBV genome integration so crucial for oncogenesis? I do not think so, because there are a lot of EBV-positive cancers without integration. It may play a role, but only partial. 

Reviewer 2 Report

While the review addresses an extremely interesting topic, the role of the tumor microenvironment in the pathogenesis of Epstein Barr virus, it is very difficult to follow in its current form and falls short of the expected benefit of a review. The English language/grammar should definitely be revised, especially the use of commas. Some sentences are difficult to understand. The authors are often imprecise and do not substantiate their statements with the scientific findings, so the reader is often left confused rather than informed.

Some chapters of the review should be reconsidered according to the focus of the review, should be worded more stringently, This concerns overall the first part of the review the transfer of background knowledge about EBV, infection cycle, transcription patterns and latency types. These chapters should be made more stringent, especially more comprehensible. The chapter 3 regarding diagnostics, on the other hand, can be omitted completely. The figures would have to be adapted. For example, Figure 1 is not of good quality and is also oversized. Furthermore, in Figure 1 EBER-ISH in latency type 2 a panel seems to be perhaps mixed up. At least the cell density appears to be different here.

An illustration to the transcription program is missing and could thereby make the rather lengthy paragraph easier for the reader.

Other points:

- -In Table 1, the numbers referring to age of onset do not match the numbers in the text. This should definitely be adjusted

- Overall, the references provided should be checked carefully. Some references are not correct or are not the work that primarily led to the results.

- - Line 65: what does the sentence "Within the lytic program of infection, different virions are produced and can be distinguished [13]" mean? I also think the references cited here are not the best.

-          

- Chapter  4 is the more interesting one and is in line with the aim of the review. However, some parts should be improved:

- Table 3: in chromatin restructuring, "lymphoma" with its ref 111 is missing, while it is cited in the text.

- In line 269: better explain how EBNA-1 can promote genomic instability (as a transcriptional transactivator of oncogenes/induces ROS formation).

- Line 298-299: what does this sentence mean: "The gene product of BZLF1, Zta, binds to sequence motifs in viral promoter regions and sometimes even prefers methylated sequence motifs."

- Paragraph 4.2 should be restructured and not all jumbled up: Viral genome methylation, host genome methylation, histone modification. As so often in the review, it remains so superficial and vague that it adds no value to the reader.

- In line 356 paragraph 4.3, there is the following sentence that refers to paragraphs 4.3.1 to 4.3.3. "In the following part, we will focus on the main mechanisms that EBV uses to shape the TME and induce tumor cell-promoting processes, from our point of view." In this paragraph, instead of listing the mechanisms that the virus uses to shape the TME, we list differences in TME composition between EBV+ and EBV- tumor or different EBV+ pathologies. The only sentence explaining the mechanism is in lines 493-496.

- Paragraph 4.4.2 should be restructured: First, a viral protein that interferes with HLA class I is described. This is followed by a description of miRNA and, in the same sentence, again a description of a viral protein, one of which has been explained previously (BILF1 lines 550 and 562).

- Figure 2 is confusing. I would suggest splitting the figure: one showing the TME of EBV-infected cells in general, and a second figure showing more specifically which miR/viral genes are involved in the various immune escape processes.

- What are EBNA 7 and 8?

- Lines 557-559 state, "In addition, upregulation of human miRs targeting components involved in the HLA class I antigen presentation pathway, so-called immunomodulatory miRs, has been identified." Ref 238 mentions EBV only in the context of the viral miR-BARTa-5p. Ref 239 refers to miR-200a-5p in general. If the authors want to include this sentence, they should better explain its meaning. Are the same human miRNAs upregulated by EVB infection? If yes, please add the correct citation.

- Table 4: The different cells (CD56+; FoxP3+...) are not explained anywhere. The markers used to define the different cell populations should be explained. In addition, the table description should include an explanation of the abbreviations.

- Lines 576-577 and 581-582: explain which cells are the inhibitory receptor. On NK cells? What is the consequence of this binding?

- Sentence in lines 645-649 does not fit here.

Other points that need to be corrected:

- Line 21: ...extrinsic evasion immune evasion.....double word.

- There are some references that are completely incorrect:

Ref. 14 (line 68 and line 776): Electron Diffraction and High-Resolution Imaging on Highly-Crystalline Beta-Chitin Microfibril.

Ref 236 (line 581 and line 1418): CDNA and genomic DNA sequence of the 21.3 KDa subunit of NADH:ubiquinone reductase (complex I) from Neurospora Crassa.

 Ref 86 (line 952): add the correct title.

Line 285 the reference to the paragraph is wrong: instead of 3.3 and 3.4, change with 4.3 and 4.4

            line 440: the reference to the paragraph is incorrect: instead of 3.5, please 4.5.

Reviewer 3 Report

This is a very extensive review on EBV in cancers, with a good and comprehensive explanation of how the microenvironment is affected. I am impressed with how complete the review is!

3 small remarks:

  1. I would prefer titles above Tables and not the legend type text that are used now. That is only necessary to change for tables 1, 2, 3 and 5.
  2. Line 221: any constellation was found: it is not clear what this means not any, no, is constellation the right word? I can not figure out what this means!
  3. Figure 1: the immunohistochemistry for LMP1 and EBNA-2 is not very clear: is it found in only single cells and is that sufficient to call the process positive?? Maybe better examples??